# Impact of COVID-19, Political, and Financial Events on the Performance of Commercial Banking Sector

Ghulam Ghouse [1],*, Muhammad Ishaq Bhatti [2,3] and Muhammad Hassam Shahid [1]

1   Economics Department, University of Lahore, Lahore 55150, Pakistan; hassam.shahid@econ.uol.edu.pk
2   S P Jain School of Global Management, Lidcombe, NSW 2141, Australia; i.bhatti@latrobe.edu.au
3   La Trobe Business School, La Trobe University, Melbourne, VIC 3083, Australia
*   Correspondence: ghulam.ghouse@econ.uol.edu.pk

**Abstract:** This paper employs a structural empirical model to gauge the possible effects of COVID-19, political and financial events on the returns and volatility of commercial banks. It observes that insured and run-prone uninsured depositors choose between differentiated commercial banks, which appears to be significantly impacted from the present pandemic, especially for the case of Pakistan's commercial banking sector. The estimated volatility series for commercial banks is measured through the GARCH model, which explains the current financial and political distress for the case of shocks from COVID-19. We calibrate by Impulse Indicator Saturation (IIS) to detect the structural breaks formed by these events in the returns and volatility series of commercial banks. It is observed that the calibrated model possesses almost all financial events that have had a prominent impact on the returns and volatility series whereas two out of eighteen political events are unimpacted.

**Keywords:** COVID-19; financial; political; PSX; commercial banks; volatility; GARCH; IIS procedure

## 1. Introduction

The country's stock markets have a substantial part in financial markets and foster industrial growth through optimum channelization of the funds between suppliers and users of funds. This is the crucial reason that the government, industries, investors, and banks are keenly observing the performance of stock markets.

The strength of the financial system, particularly the banking sector, is a significant part of the infrastructure for robust monetary and macroeconomic policy performance on the national level (Javaid et al. 2011). The banking sector plays a significant role in accumulating funds from the savers and providing them to investors. They are offering their roles as mediators between the savers and investors (Opoku-Agyemang 2015). The financial sector permits enormous investment and efficient capital allocation through a banking system, leading to income growth (Claessens and Feijen 2007).

The financial sector of Pakistan mainly consists of the banking sector, and its size is almost three-fourth of the financial sector of Pakistan (Fazl-E-Haider 2018). Hamza and Khan (2014) explain that the banking sector in Pakistan has been a leading institution and rapidly growing for the last two decades. Pakistani Banks offers extensive facilities and services to cater clients need for conventional as well as Islamic Finance[1] including online Ijar facilities (see Adeinat et al. 2019).

The stock market's stable performance and development appeal to foreign and domestic investors and significant effects on the economic competitiveness of the country (Levine and Zervos 1998). The prevailing debate is that the stock market is susceptible to significant political, financial, and natural events such as COVID-19 that can indirectly correlate to a firm's performance. These events may generate some disturbance, affecting stock market performance (Javid and Ahmad 2020).

The total market capitalization of commercial banks in the Pakistan Stock Exchange (PSX) was 1286.23 PKR billion in 2020 (17% of the total market capitalization). The total

share of commercial banks' capitalization is 24% of the total market capitalization of the KSE 100 index. There are 20 commercial banks registered in PSX, and only 11 banks are enlisted in KSE 100 index (PSX 2019).

The figures, as mentioned earlier, show the commercial sector has a significant percentage in PSX, revealing that any adverse shock in PSX can distress the commercial sector. Various literature has been carried out on the impact of party-political actions on the PSX (Javed and Ahmed 1999; Ahmad et al. 2017; Nazir et al. 2014; Mahmood et al. 2014; Murtaza and Ali 2015; Ghouse et al. 2019). Furthermore, the studies also discovered the effect of an economic event on the performance of the Pakistan stock exchange. Several studies checked the impact of both domestic and foreign economic events on PSX variation (Sohail and Javid 2014; Nazir et al. 2010).

No such study notably identified the structural breaks due to COVID-19, financial and political events on commercial banks' stock prices. In comparison, identifying structural breaks is very important because any break in series could inflate the variation in modeling outputs and forecasting. These errors make the inferences of the model and forecasting invalid. Plenty of studies used different modeling such as GARCH modeling, Event Window Analysis, but in this study, we used Impulse Indicator Saturation which is not used for similar studies. Therefore, the main objective of this study is to analyze the influence of some significant COVID-19 financial and political events on commercial banks. Moreover, this study also explores the influence of these events on the return series of stock prices of commercial banks. Furthermore, the present study also captures the variation in commercial banks' stock prices.

The current research picks three commercial banks; "Habib Bank Limited (HBL), Muslim Commercial Bank (MCB), and United Bank Limited (UBL)", which are listed in KSE 100 index. This study covered the financial and political events from October 2007 to July 2020. This study selects only those events, that prominently hit the stock market causing high fluctuations in the data period as detailed in Table 1 below.

**Table 1.** The events and its sources.

| Dates | Category | Detail | Source |
|-------|----------|--------|--------|
| **14-November-07** | Internal Affair | Closure of TV Channels | http://www.brecorder.com/2007/11/20071120 654385/ (accessed on 10 August 2021) |
| **19-February-08** | Pakistan's General Elections 2008 | PPP elected their Prime Minister | https://www.reuters.com/article/us-pakistan-politics-idUSISL7451920080324 (accessed on 10 August 2021) |
| **15-May-08** | Internal Affair | Lawyers Movement for Chief Justice of Pakistan (Justice Iftikhar Ahmed) | https://www.nytimes.com/2008/06/01//01 PAKISTAN-t.html (accessed on 10 August 2021) |
| **7-September-08** | The Financial crisis | 2008 Global Financial crisis | https://www.federalreservehistory.org/essays/ great-recession-and-its-aftermath (accessed on 10 August 2021) |
| **11-March-09** | Internal Affair | Long March organized by PML-N | https://www.aljazeera.com/news/2009/3/12/ pakistans-long-march-to-stability (accessed on 10 August 2021) |
| **10-June-10** | Internal Affair | Punjab Government opposed Rah-e-Najaat | https: //www.pakistanarmy.gov.pk/WOT-year.php (accessed on 10 August 2021) |
| **2-May-11** | Foreign Affair | Operation Neptune Spear | https: //www.911memorial.org/learn/resources/ digital-exhibitions/digital-exhibition-revealed-hunt-bin-laden/operation-neptune-spear (accessed on 10 August 2021) |

**Table 1.** *Cont.*

| Dates | Category | Detail | Source |
|---|---|---|---|
| 28-October-11 | Long March/Political Gathering | Long March started by PMLN | https://onlinelibrary.wiley.com/doi/abs/10.1111/j.2041-9066.2011.00068.x (accessed on 10 August 2021) |
| 31-October-11 | Political Gathering | Pakistan Tehreek-e-Insaf organized a political gathering | https://www.latimes.com/world/la-xpm-2011-dec-26-la-fg-pakistan-rally-20111226-story.html (accessed on 10 August 2021) |
| 18-June-12 | Internal Affair | Yousef Raza Gillani dismissal | https://www.dawn.com/news/727782 (accessed on 10 August 2021) |
| 15-January-13 | Long March/Political Gathering | Pakistan Awami Tehreek Protest against PPP | https://www.dw.com/en/no-revolution-in-pakistan-at-least-not-for-now/a-16532257 (accessed on 10 August 2021) |
| 13-May-13 | Elections | PML-N formed a new Government in Pakistan | https://www.reuters.com/article/us-pakistan-election-idUSBRE94F0B520130516 (accessed on 10 August 2021) |
| 5-June-13 | Internal Affair | Nawaz Sharif became the 20th PM of Pakistan | https://na.gov.pk/en/priminister_list.php (accessed on 10 August 2021) |
| 15-August-14 | Long March/Political Gathering | Protest against rigging by PAT and PTI | https://participedia.net/case/4514 (accessed on 10 August 2021) |
| 19-August-14 | Internal Affair | Announcement of Civil disobedience by PTI | https://www.dawn.com/news/1126126 (accessed on 10 August 2021) |
| 20-February-18 | Internal Affair | Supreme Court terminated Nawaz Sharif on embalmment | https://www.dawn.com/news/1348191 (accessed on 10 August 2021) |
| 13-July-18 | Internal Affair | PML-N leaders arrested by Police | https://www.bbc.com/news/world-asia-49277829 (accessed on 10 August 2021) |
| 25-July-18 | Internal Affair | New Elections conducted in Pakistan | https://www.orfonline.org/research/pakistan-general-elections-2018-analysis-of-results-and-implications-46324/ (accessed on 10 August 2021) |
| 26-July-18 | Internal Affair | PTI bagged the highest seat | https://www.ecp.gov.pk/frmGenericPage.aspx?PageID=3213 (accessed on 10 August 2021) |
| 9-October-18 | Exchange rate | Pakaistani rupee depreciation | https://www.dawn.com/news/1651372 (accessed on 10 August 2021) |
| 30-November-18 | Exchange rate | Rupee depreciation from 131.95 to 136.51 | https://www.cnbc.com/2018/11/30/pakistani-rupee-plunges-in-likely-central-bank-devaluation-traders.html (accessed on 10 August 2021) |
| 19-March-20 | The first outbreak of COVID-19 | First COVID-19 patient admitted to a hospital. | http://www.emro.who.int/pak/pakistan-news/who-extends-support-to-pakistan-as-it-confirms-its-first-two-cases-of-covid-19.html (accessed on 10 August 2021) |
| 23-March-20 | First, lockdown in Karachi | Lockdown in Karachi and closure of PSX. | https://www.aa.com.tr/en/asia-pacific/pakistan-karachis-province-set-for-covid-19-lockdown/1774971 (accessed on 10 August 2021) |
| 31-May-20 | Nation-wide lockdown due to COVID-19 | Nation-wide lockdown due to COVID-19. | https://crisis24.garda.com/alerts/2020/04/pakistan-government-extends-nationwide-lockdown-until-april-30-update-18?origin=fr_riskalert (accessed on 10 August 2021) |

## 2. Literature Review

Literature review revealed that many studies have been carried out on the impact of political events on the stock exchange. Primarily these studies are conducted to analyze the impact of the USA presidential elections on stock markets (Niederhoffer et al. 1970; Herbst and Slinkman 1984; Hobbs and Riley 1984; Gärtner and Wellershoff 1995). Im et al. (2001) examined the effect of the USA and Canadian government elections on Canadian stocks. Others who researched in this area are Foerster and Schmitz (1997) who supported and confirmed the finding of (Im et al. 2001). Recently, Pantzalis et al. (2000) conducted a study that explored the influence of political instabilities on different stock markets. Few concessions are observed in this study, but the results are mixed overall. Some studies found the stock markets get upward trends in the republican government (Riley and Luksetich 1980; Hobbs and Riley 1984). In contrast, (Huang 1985; Gärtner and Wellershoff 1995) showed that stock returns did not systematically fluctuate during democratic and republican governments.

The studies reviewed above did not cover the type of government system; democratic or autocratic. In a two-party structure with more votes and consequently single-party governments, election covered all ambiguities regarding future policies. Most governments are in power based on an alliance with other countries. This suggests lag reaction of the stock returns and election. Moreover, commonly the governments are unstable and always tend to fall, and a new government could be made without an election. Ultimately these structural changes affect the performance of stock markets (Ghouse et al. 2021b; Irshad 2017).

The financial crisis of 2007–2008 occurred because lending led to an economic turmoil, causing economies to face market fall. In 2007, stock prices of SandP500 index fell almost 50%, and the economies got back to a GDP of the 1990s. For instance, the GDP of Japan dropped around 15% in the first quarter of 2009 (Jones and Banning 2009). Another study by Neaime (2012) investigated the impact of 2008 financial crises, which studies explicitly how the stock prices of North Africa were shrunk. As per the results, economies with a comparatively immense amount of economic trade with the USA and Europe appeared to be more affected by the financial crises. Dimpfl (2011) observed the influence of news on the stock market in Germany, and found that Frankfurt Stock Exchange DAX prices are sensitive to the news. As indicated above, if the information is publicized in the market, the outcome on prices can be seen immediately (Fama et al. 1969). Likewise, Chen et al. (1986) also examined the association between stock prices and economic events. He found that the changes in stock prices are due to the variation in inflation. He concluded that markets are vulnerable to the news in an economy.

Narayan and Narayan (2012) explicitly investigated the effect of major macroeconomic events from 2000–2010 on Asian stock markets. The results are indicated that some stock markets sensitively respond to events and few markets did not respond in the same fashion. The Asian market responded to the USA central bank policy changes (Kim 2009). However, Pearce and Roley (1984) claimed that every economic events did not affect stock prices. They explored the impact of economic news on the financial market. The results indicated that there is weak evidence that news related to economic activity did not affect the stock prices.

Currently, the coronavirus pandemic continues to spread worldwide, affecting economies and financial market, and causing imbalances and instability in economies. Baker et al. (2020) exposed that in the last three weeks, 20 stock markets nosedived, due to the "bad news" of COVID-19. Likewise, in the same period, the USA economy also faced a severe challenge. The COVID-19 consequences are frequently compared with the 2008 financial crisis (Kenourgios et al. 2011; Coatesworth and Dimitriou 2013; Bekiros 2014; Luchtenberg and Vu 2015; Yarovaya et al. 2016). However, in the COVID-19 crises, the distinct catastrophe shock is apparent, which is an extent of COVID-19. Kraus et al. (2020) emphasized on the implications of global financial crises and the coronavirus outbreak.

Many countries implicated that the lockdown badly affected the businesses, job market, and essential services. Few argued that the inquiry of financial markets in reaction to COVID-19 is of great importance (Zhang et al. 2020) to supply-demand analysis in the labor market (Patnaik 2022). Conversely, we claim that the coronavirus pandemic is a kind

of crisis that originates from a different contagion and is comparable to both the global financial crisis and world wars. A more precise and perfect comparison made with the pandemic and likewise events are presented in (Lilley et al. 2020; Eichenbaum et al. 2020; He et al. 2020). One month after the start of the COVID-19 pandemic from China, a sudden 30% fall in oil prices has been observed (Norouzi et al. 2020). These severe shocks have provoked a downward trend in the stock markets. Aloui et al. (2011) argued that the economies with greater sensitivity to oil price variation tend to co-move with the USA economy. Ho and Gan (2021) explored the effect of pandemic on the FDI. They use world pandemic uncertainty index. They concluded that the health pandemics cause negative impact on FDI.

The COVID-19 epidemic is a cause of risk and fluctuations which is why more advanced research are needed to analyze these effects. Therefore, this study explores the impact of COVID-19 financial and political events on the commercial banking sector of Pakistan stock market. Pakistan's market is sensitive to natural, economic and political events and perceives spillover effect from other stock markets. Second, the extension of COVID-19 in Pakistan has generated a catastrophic disaster in China, Italy, the USA, and Iran.

## 3. Methodology and Model Specifications

It is indispensable to choose suitable or relevant measures to evaluate the model through appropriate econometric model to meet empirical policy guidelines and objectives. The daily data are used from the period of October 2007 to July 2020. The data set contains stock prices of the top three commercial banks, HBL, MCB, and UBL, listed in KSE 100. Stock prices data are collected from the Business recorder, political and financial events data are collected from DAWN News, Tribune News, Geo News, KSE, Express Newspaper, Global issues, Guardian, Pakistan Today, Defense.pk, HRW, BBC, CNBC, and Election Commission of Pakistan (ECP).

Numerous studies use the OLS technique to investigate the impact of an event on stock prices. The most critical assumption of the ordinary least square (OLS) model is that the variances are not time-varying. However, we cannot apply the OLS model to capture the relationship when they are not homogenous. When the series contains time-varying variance, we commonly use the ARCH model (Engle 1982). It is well established that financial series have an ARCH effect. We use ARCH modelling instead of OLS model, due to the presence of ARCH in our series. However, the GARCH model proved to be a notable inclusion to financial literature and ARCH model. GARCH model is also used to compute volatility series of banks.

Moreover, the IIS procedure has been applied to check the impact of political and financial events. The IIS procedure is employed for the deification of structural breaks. The Impulse Indicator Saturation IIS captures the changes in the series level and multiple breaks (Doornik et al. 2013).

### 3.1. Model Specifications

The financial series are usually trendy; that is why it is not possible to get robust and unbiased results from these series without dealing with a trend (Ghouse and Khan 2017; Ghouse et al. 2019; Ghouse et al. 2021a). To tackle this issue, we applied the log difference of the financial series. Where the difference de-trended the series and log reduced the dispersion around the mean value of the series. The method of log differencing is the following:

$$R_t = \log\left(\frac{p_t}{p_{t-1}}\right) \tag{1}$$

The $p_t$ is the current price, i.e., the stock price at time t and $p_{t-1}$ is the lag price of the stock series.

### 3.1.1. ARCH Model

Engle pioneered the Autoregressive Conditional Heteroscedastic (ARCH) procedure (Engle 1982). The ARCH model deals with two equations at the same time. The first equation finds out the data-generating process of the conditional mean equation, and the second model the conditional variance equation. The generalized form of equations of the ARCH model is presented in the below-given equation.

Conditional Mean Equation:

$$R_t = \pi_0 + \pi_1 G_t + \varepsilon_t \tag{2}$$

where $\varepsilon_t = z_t \sigma_t$, $z_t \sim N(0,1)$.

Conditional Variance Equation:

$$\sigma_t^2 = \delta_0 + \sum_{i=1}^{q} \delta_i \varepsilon_{t-i}^2 + u_t \tag{3}$$

where, $i = 1, 2, \ldots, q$.

The $R_t$ shows the return series and $\pi_1$ indicates the vector of the parameters of the ARMA procedure. $\pi_1 G_t$ indicates the generalized method of the ARMA (p,q) procedure. The ARCH model has some basic restrictions, as the parameter's sign of conditional variance equation must be positive. The ARCH model only captures the effect when the effect is symmetric. The $\varepsilon_t$ shows disturbance and $\varepsilon_{t-1}^2$ is measured as an ARCH term.

### 3.1.2. GARCH Model

Nonetheless, GARCH model minimized the long lag length problem of ARCH model while creating an ease in the degree of freedom. Therefore, to solve this issue, Bollerslev (1986) presented the GARCH model. Ghouse and Khan (2017) used the GARCH modeling to model the volatility of stock prices of different markets. Sohail and Javid (2014) applied GARCH modeling to measure inflation and its uncertainty in Pakistan. Ghouse et al. (2021b) used GARCH model for volatility modeling and spillover effects. The GARCH model differs from the ARCH specification by taking the lagged values of the conditional variance equation as an independent variable. The general representation of the GARCH (p,q) model is the following:

Conditional Mean Equation:

$$R_t = \pi_0 + \pi_1 G_t + \varepsilon_t \tag{4}$$

where $\varepsilon_t = z_t \sigma_t$, $z_t \sim N(0,1)$.

Conditional Mean Equation:

$$\sigma_t^2 = \delta_0 + \sum_{i=1}^{q} \delta_i \varepsilon_{t-i}^2 + \sum_{j=1}^{p} \vartheta_j \sigma_{t-j}^2 + u_t \varepsilon \tag{5}$$

The $R_t$ displays the returns and $\pi_1$ indicates the vector of the parameters of the ARMA process. The GARCH model only captures the effect when the effect is symmetric. $\sigma_{t-j}^2$ denotes the lagged value of conditional variance. The $\vartheta_j$ is the vector parameter of $\sigma_{t-j}^2$. While $\delta_i$ is the vector parameter of ARCH $\varepsilon_{t-i}^2$ effect.

### 3.1.3. Impulse Indicator Saturation (IIS)

Santos et al. (2008) introduced the Impulse Indicator Saturation (IIS) procedure. The essential tenacity of this procedure is to identify the intercept shift, breaks, multiple breaks, and co-breaks. We generate a generalized unrestricted model (GUM) through this procedure. This technique generates dummies equal to observations, which means each observation has its binary dummy variable. This way, it checks the change or breaks at every point of data. Simply, it seems that the IIS violates the "degree of freedom" rule. However,

technically, it works by considering this assumption by introducing a specific number of dummies in the first regression and remaining dummies in other regressions, confirming that this procedure divided the dummies into many parts and ran many regressions. In this way, the problem of over-parameterization can be solved. Therefore, this technique also sets the significance level at which we want to see the breaks or shifts. This analysis aims to explore the significant impact of both political and financial events on commercial banks. Therefore, we set the level of significance at 5%. Suppose the dummies are significant at the exact date and following dates when the event occurred, in that case, this event has a significant impact on the series, and we recheck their joint significance. The equations of the Impulse Indicator Saturation (IIS) procedure are the following:

$$R_{it} = \delta_0 + \delta_1 R_{it-1} + \sum_{t1=1}^{250} \gamma_t D_{it} + \varepsilon_{it} \tag{6}$$

$$\varepsilon_{it} \sim \text{IIN}\left(0, \sigma_t^2\right) t_1 = 1, 2, \ldots, 250$$

$$R_{i,t} = \delta_0 + \delta_1 R_{it-1} + \sum_{t1=251}^{500} \gamma_{t2} D_{i, t} + \varepsilon_{i,t} \tag{7}$$

$$\varepsilon_{it} \sim \text{IIN}\left(0, \sigma_t^2\right) t_2 = 251, 252, \ldots, 500$$

and the last equation is the following:

$$R_{i,t} = \delta_0 + \delta_1 R_{it-1} + \sum_{t1=251}^{500} \gamma_{t2} D_{i, t} + \varepsilon_{i,t} \tag{8}$$

$$\varepsilon_{it} \sim \text{IIN}\left(0, \sigma_t^2\right) t_2 = 251, 252, \ldots, 500$$

The data set splits our data into 12 parts by introducing 250 dummies in each regression. We can see that $t_1$ shows that it varies from 1 to 250 and then $t_2$ varies from 251 to 500 and so on $t_{12}$ varies from 2751 to 2780.

## 4. Results and Discussion

This section discusses the empirical results and their theoretically possible interpretation. First, the visualization of the series is carried out to understand the characteristics of the series. Second, the GARCH model is used to estimate the volatility of the series. Third, IIS is employed to check the significant shifts of politics, financial, and COVID-19 occasions in the stock price series of banks.

### 4.1. Graphical Analysis

The data visualization apprehends the dynamics and behavior of the stock price series of "Habib Bank Limited, Muslim Commercial Bank, and United Bank Limited". Figure 1 displays the series has a rising (downward) movement with huge fluctuations.

These series move downward initially, which is most probably due to the global financial crisis of 2008. After that, all series attained an upward trend, which is a sign of recovery but with large volatilities due to the impact of certain financial, political, and natural events. The series shows downward trends again in the era of COVID-19.

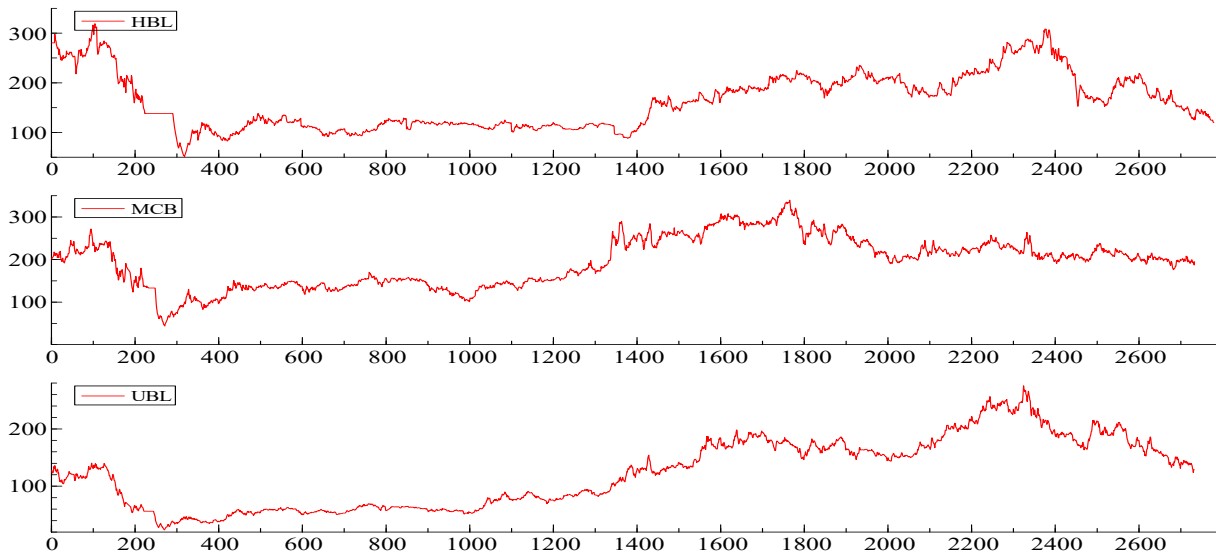

**Figure 1.** The raw series of HBL, MCB, and UBL.

Figure 2 shows the return series of stock prices attained after taking log difference. This series shows high and low fluctuation. These bunches of fluctuations generated an autoregressive conditional heteroscedastic (ARCH) effect. These fluctuations are also termed volatility, and the low volatility shows less impact of an event, and high volatility indicates the massive impact of an event on return series. The return series of commercial banks also shows fluctuations due to COVID-19.

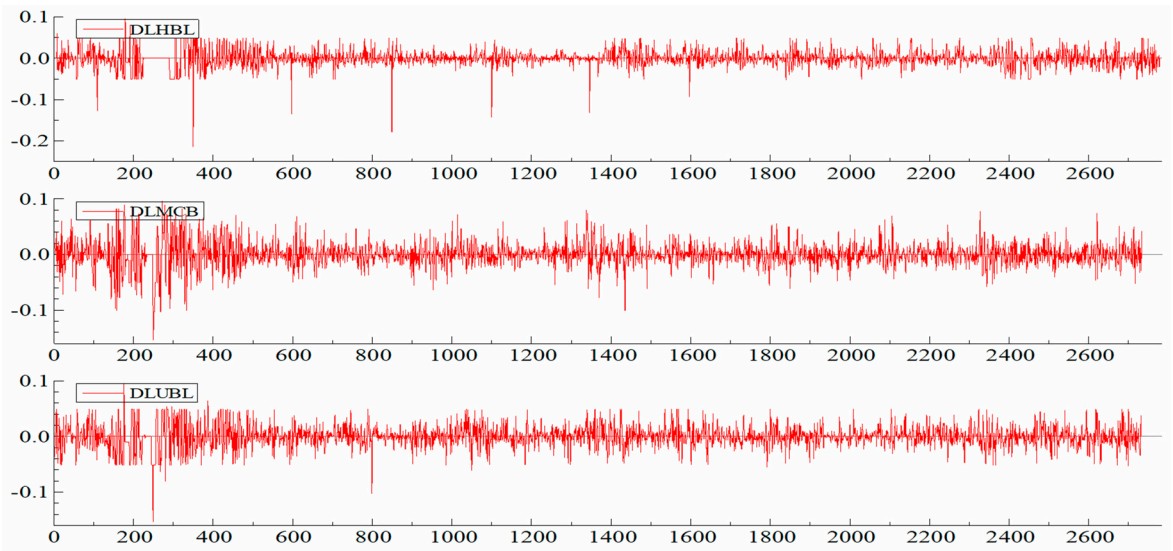

**Figure 2.** The return series of HBL, MCB, and UBL.

### 4.2. Descriptive Statistics

Basic descriptive statistics show the primary statistics of the series, which provide basic knowledge about the nature of the series. Below given Table 2 describes the summary of statistics of HBL, MCB, and UBL returns. The mean of these series is negative and moving around zero value, demonstrating that the series has mean reversion behavior. The skew-ness is concerned about the symmetry of the distribution's tails, and the statistics show that the series is negatively skewed. It means there are outliers on the negative distribution. The Kurtosis is related to the peak of the distribution. The figures illustrate that the series is leptokurtic, which means the series has a higher peak than a normal

reference distribution. The Jarque-Bera test is apprehended to check the normality, and its stats show that the series follows the normal distribution.

**Table 2.** Descriptive statistics of HBL, MCB, and UBL return series.

| Series | Mean | Standard Deviation | Skewness | Jarque Bera | Excess Kurtosis | Q-Stat (5) | Q2-Stat (5) | ARCH 1–2 | KPSS |
|---|---|---|---|---|---|---|---|---|---|
| HBL | −0.2149 | 0.0204 | −0.9096 (0.0000) | 1168 (0.0000) | 9.968 (0.0000) | 79.365 (0.0000) | 89.703 (0.0000) | 25.503 (0.0000) | 0.199 |
| MCB | −0.1539 | 0.0226 | −0.1889 (0.0000) | 1517.5 (0.0021) | 3.632 (0.0000) | 23.595 (0.0000) | 23.157 (0.0000) | 4.849 (0.0003) | 0.072 |
| UBL | −0.1538 | 0.0200 | −0.1838 (0.0000) | 650.48 (0.0000) | 2.363 (0.0000) | 4.3332 (0.5025) | 6.766 (0.9290) | 126.86 (0.0000) | 0.245 |

**Hypotheses**
"KPSS H0: Return series is level stationary; asymptotic significant values 1% (0.739), 5% (0.463), 10% (0.347). Q-stat (return series) there is no serial autocorrelation. Q2-stat (square return series) H0: there is no serial autocorrelation. Jarque-Bera H0: distribution of series is normal. LM-ARCH H0: there is no ARCH effect. Use these asymptotic significance values of *t*-stat 1% (0.01), 5% (0.05), 10% (0.1) and compare these critical values with *p*-values (Probability values). *p*-values are in the parenthesis."

The ARCH test statistics show that series have an ARCH effect. Moreover, the "KPSS test" is applied to find the unit root, and the results show that all series are stationary.

### 4.3. Volatility Modeling

To find out the return's volatility, we used the GARCH model because the ARCH effect is present in these series. Table 3 displays the HBL's GARCH results. The first section of below given table shows the conditional mean equation results that describe that the returns follow moving average behavior. That is the reason the autoregressive (AR) term is zero in this model. In the second section, the conditional variance equation's results refer that the ARCH and GARCH terms are significant in the model.

**Table 3.** The results of GARCH model for HBL return series.

| Parameters | Coefficient | Standard Error | *t*-Value | *t*-Probability |
|---|---|---|---|---|
| | | **Conditional Mean Equation** | | |
| **Constant** $\theta_0$ | 0.0078 * | 0.0001 | 0.3419 | 0.0732 |
| **AR(1)** $\vartheta_1$ | - | - | - | - |
| **MA(1)** $\varnothing_1$ | 0.0767 *** | 0.0231 | 3.3250 | 0.0009 |
| | | **Conditional Variance Equation** | | |
| **Constant** $\gamma_0$ | 0.0056 * | 0.0000 | 0.2181 | 0.0827 |
| **ARCH(1)** $\gamma_1$ | 0.3079 *** | 0.0270 | 11.4000 | 0.0000 |
| **GARCH(1)** $\delta_1$ | 0.8051 *** | 0.0388 | 20.7600 | 0.0000 |
| **Student (DF)** | 3.0034 *** | 0.3337 | 9.0010 | 0.0000 |
| **Persistence of shock** | | 0.9655 | | |

**Hypotheses for nth order**
"AR (p) H0: $\vartheta_i = 0$ No AR Process, MA (q) H0: $\varnothing_i = 0$ No MA Process, ARCH H0: $\theta_i = 0$ No ARCH effect, GARCH H0: $\varphi_i = 0$ No GARCH effect. The * and *** are showing the significance at 10% and 1% respectively".

| | | | **Residual Analysis** | | | | |
|---|---|---|---|---|---|---|---|
| Tests | Jarque Bera | Q-Stat (5) | Q-Stat (10) | Q²-Stat (5) | Q²-Stat (10) | LM-ARCH (1-2) | LM-ARCH (1–5) |
| Values | 7.0768 *** (0.0000) | 1.8970 (0.9988) | 0.4549 (0.8476) | 0.9023 (0.9780) | 0.3134 * (0.0990) | 0.5654 (0.9897) | 0.7843 (0.9340) |

**Hypotheses for nth order**
"Q-stat (return series) there is no serial autocorrelation. Q²-stat (square return series) H0: there is no serial autocorrelation. Jarque-Bera H0: distribution of series is normal. LM-ARCH $H_0$: there is no ARCH effect. *p*-values are in the parenthesis".

The student "*t*" distribution parameter is significant in the model, which illustrates that the HBL returns distribution is not normal. The persistence of the shock parameter is close to one, which reflects that the ARCH and GARCH impacts take a longer time to decline. The next section of the table demonstrates the residual analysis. The results of this test expound that the test statistic is insignificant, which means there is no autocorrelation at 5th and 10th lags. The Q-square stat test is also used for autocorrelation in residuals variances. The results illustrate that the test statistic is insignificant, meaning no heteroscedasticity at 5th and 10th lags. The LM-ARCH test results also show no ARCH effect, which means the model is a good fit.

Table 4 given below presents the GARCH model's result for MCB. In the first section of Table 4, the results of the conditional mean equation define that the returns follow moving average and autoregressive behavior. That is why the AR and MA parameters are significant in this model. The next part of the table highlights the conditional variance equation results denoting that the ARCH and GARCH terms are significant in the model.

**Table 4.** The results of GARCH model for MCB return series.

| Parameters | Coefficient | Standard Error | *t*-Value | *t*-Probability |
|---|---|---|---|---|
| **Conditional Mean Equation** | | | | |
| Constant $\theta_0$ | 0.0000 | 0.0002 | −0.0054 | 0.9957 |
| AR (1) $\vartheta_1$ | 0.9137 *** | 0.1292 | 7.0700 | 0.0000 |
| MA (1) $\varnothing_1$ | −0.9295 *** | 0.1305 | −7.1230 | 0.0000 |
| **Conditional Variance Equation** | | | | |
| Constant $\gamma_0$ | 0.1923 *** | 0.0469 | 4.0960 | 0.0000 |
| ARCH (1) $\gamma_1$ | 0.1843 *** | 0.0282 | 6.5310 | 0.0000 |
| GARCH (1) $\delta_1$ | 0.7974 *** | 0.0275 | 29.0100 | 0.0000 |
| Student (DF) | 4.6308 *** | 0.4241 | 10.9200 | 0.0000 |
| Persistence of shock | | 0.9840 | | |

**Hypotheses for nth order**
"AR (p) H0: $\vartheta_i = 0$ No AR Process, MA (q) H0: $\varnothing_i = 0$ No MA Process, ARCH H0: $\theta_i = 0$ No ARCH effect, GARCH H0: $\varphi_i = 0$ No GARCH effect. The *** is showing the significance at 1%".

| | | | **Residual Analysis** | | | | |
|---|---|---|---|---|---|---|---|
| Tests | Jarque Bera | Q-Stat (5) | Q-Stat (10) | $Q^2$-Stat (5) | $Q^2$-Stat (10) | LM-ARCH (1-2) | LM-ARCH (1–5) |
| Values | 6.8681 (0.0000) | 2.3071 (0.9999) | 0.6750 (0.1970) | 0.7898 (0.9882) | 0.3741 (0.9990) | 0.5508 (0.4509) | 0.8702 (0.5670) |

**Hypotheses for nth order**
"Q-stat (return series) there is no serial autocorrelation. $Q^2$-stat (square return series) H0: there is no serial autocorrelation. Jarque-Bera H0: distribution of series is normal. LM-ARCH $H_0$: there is no ARCH effect. *p*-values are in the parenthesis."

The student "*t*" distribution parameter is significant in the GARCH model, which shows that the MCB returns distribution is not normal. The persistence of the shock parameter is 0.9840, which is close to one, which illustrates that both the ARCH and GARCH impacts take more time to decrease. The third panel illustrates the result of the residual analysis. The Q-stat test results explain that the test statistic is insignificant, meaning no autocorrelation at 5th and 10th lags. The Q-square stat test results explain that the test statistic is insignificant, meaning no heteroscedasticity at 5th and 10th lags. The LM-ARCH test results are also showing that there is no ARCH effect.

Table 5 expresses the UBL GARCH model results. First part of table shows the results of the conditional mean equation describe that the returns follow autoregressive behavior but not moving average. That is why the MA term is zero in this model. In the second

section, the conditional variance equation results refer that the ARCH and GARCH terms are significant in the model.

**Table 5.** The results of GARCH model for UBL return series.

| Parameters | Coefficient | Standard Error | *t*-Value | *t*-Probability |
|---|---|---|---|---|
| **Conditional Mean Equation** | | | | |
| Constant $\theta_0$ | 0.0000 | 0.0003 | 0.0322 | 0.9743 |
| AR (1) $\vartheta_1$ | 0.0833 *** | 0.0209 | 3.9960 | 0.0001 |
| MA (1) $\varnothing_1$ | - | - | - | - |
| **Conditional Variance Equation** | | | | |
| Constant $\gamma_0$ | 0.1575 *** | 0.0414 | 3.8030 | 0.0001 |
| ARCH (1) $\gamma_1$ | 0.1792 *** | 0.0292 | 6.1330 | 0.0000 |
| GARCH (1) $\delta_1$ | 0.8022 *** | 0.0286 | 28.0400 | 0.0000 |
| Student (DF) | 5.2187 *** | 0.5915 | 8.8230 | 0.0000 |
| **Persistence of shock** | | 0.9814 | | |

**Hypotheses for nth order**
"AR (p) H0: $\vartheta_i = 0$ No AR Process, MA (q) H0: $\varnothing_i = 0$ No MA Process, ARCH H0: $\theta_i = 0$ No ARCH effect, GARCH H0: $\varphi_i = 0$ No GARCH effect. The *** is showing the significance at 1%".

| | **Residual Analysis** | | | | | | |
|---|---|---|---|---|---|---|---|
| Tests | Jarque Bera | Q-Stat (5) | Q-Stat (10) | $Q^2$-Stat (5) | $Q^2$-Stat (10) | LM-ARCH (1–2) | LM-ARCH (1–5) |
| Values | 9.7853 (0.0000) | 1.7806 (0.9999) | 0.4356 (0.7643) | 0.4520 (0.9881) | 0.3741 (0.9990) | 0.8759 (0.6742) | 0.6930 (0.3286) |

**Hypotheses for nth order**
"Q-stat (return series) there is no serial autocorrelation. $Q^2$-stat (square return series) H0: there is no serial autocorrelation. Jarque-Bera H0: distribution of series is normal. LM-ARCH $H_0$: there is no ARCH effect. *p*-values are in the parenthesis."

The last panel explains the residual analysis for the validity of the results. The Q-stat test results show that the test statistic is insignificant, reflecting no autocorrelation at 5th and 10th lags. The Q-square stat test results explain that the test statistic is insignificant, meaning there is no heteroscedasticity at 5th and 10th lags. The LM-ARCH test results also show that there is no ARCH effect.

The volatilities of the series are estimated through conditional variance equations from the entire model. The principal objective is to check the impact of political and financial events on commercial bank stock prices in the KSE 100 index. The Impulse Indicator Saturation provides *t*-stat values for each point that cannot be explained in the draft. Therefore, this study employed an F-stat test on the significant dummies and checked their joint significance.

Table 6 presents the impact of COVID-19, political and financial events on stock prices of commercial banks; HBL, MCB, and UBL. This study selects only those events, that prominently hit the stock market and cause high fluctuations in the data period, for details see Table 1. The results in Table 6 indicate that the COVID-19 badly affected the performance of commercial banking sector. The COVID-19 outbreak impacted the performance of stock market (Ahmed 2020). The lockdown in Karachi and the nation-wide lockdown caused poor performance of Pakistan stock market (Kalsoom et al. 2021; Waheed et al. 2020), they also influenced Islamic banking sector (Ghouse et al. 2021b). All the events that occur due to COVID-19 have significant impact stock prices of commercial banks and bring down the performance of these banks.

**Table 6.** The results of impact of events on returns of banks.

| Date | Category | Detail | HBL (F-Stat) | MCB (F-Stat) | UBL (F-Stat) |
|---|---|---|---|---|---|
| 14-November-07 | Internal Affair | Closure of TV Channels | 2.6140 0.2385 | 1.6022 0.6239 | 0.7562 0.8322 |
| 19-Feburary-08 | Elections | PPP elected their Prime Minister | 180.693 *** 0.0052 | 16.944 *** 0.0016 | 76.456 *** 0.0028 |
| 15-May-08 | Internal Affair | Lawyers Movement for Chief Justice of Pakistan (Justice Iftikhar Ahmed) | 171.885 ** 0.0494 | 93.794 ** 0.0387 | 63.865 ** 0.0376 |
| 7-September-08 | financial crisis | 2008 Global Financial crisis | 148.313 *** 0.0000 | 18.639 *** 0.0000 | 64.473 *** 0.0000 |
| 11-March-09 | Internal Affair | Long March organizaed by PML-N | 42.598 *** 0.0000 | 36.342 *** 0.0000 | 39.874 *** 0.0000 |
| 10-June-10 | Internal Affair | Punjab Government opposed Rah-e-Najaat | 2.1804 0.5004 | 1.7682 0.6649 | 2.3832 0.1539 |
| 2-May-11 | Foreign Affair | Operation Neptune Spear | 111.137 *** 0.0029 | 76.541 *** 0.0007 | 85.328 *** 0.0005 |
| 28-October-11 | Political Gathering | Long March started by PMLN | 120.758 *** 0.0000 | 98.653 *** 0.0000 | 86.234 *** 0.0000 |
| 31-October-11 | Political Gathering | Pakistan Tehreek-e-Insaf organized a political gathering | 191.055 *** 0.0000 | 48.764 *** 0.0000 | 81.582 *** 0.0000 |
| 18-June-12 | Internal Affair | Yousef Raza Gillani dismissal | 33.397 *** 0.0000 | 45.327 *** 0.0000 | 25.460 *** 0.0059 |
| 15-January-13 | Long March | Pakistan Awami Tehreek Protest against PPP | 74.6131 *** 0.0039 | 65.550 *** 0.0014 | 55.557 *** 0.0001 |
| 13-May-13 | Elections | PML-N formed a new Government in Pakistan | 145.129 *** 0.0000 | 57.475 *** 0.0000 | 74.982 *** 0.0000 |
| 5-June-13 | Internal Affair | Nawaz Sharif became the 20th PM of Pakistan | 4.386 * 0.0957 | 3.998 * 0.0846 | 3.375 * 0.0627 |
| 15-August-14 | Protest | Protest against riggning by PAT and PTI | 129.176 *** 0.0000 | 92.936 *** 0.0000 | 83.235 *** 0.0060 |
| 19-August-14 | Internal Affair | Announcement of Civil disobedience by PTI | 8.910 * 0.0654 | 9.757 * 0.0721 | 9.487 * 0.0721 |
| 20-February-18 | Internal Affair | Supreme Court terminated Nawaz Sharif on embazalment | 187.767 *** 0.0000 | 86.674 *** 0.0000 | 34.876 *** 0.0000 |
| 13-July-18 | Internal Affair | PML-N leaders arrested by Police | 19.602 *** 0.0000 | 62.798 *** 0.0000 | 71.843 0.0002 |
| 25-July-18 | Internal Affair | New Elections conducted in Pakistan | 113.855 *** 0.0000 | 48.765 *** 0.0000 | 41.532 *** 0.0000 |
| 26-July-18 | Internal Affair | PTI bagged highest seat | 143.065 * 0.0729 | 56.102 * 0.0694 | 68.385 * 0.0694 |
| 9-October-18 | Exchange rate | Pakaistani rupee depreciation | 58.530 *** 0.0000 | 86.876 *** 0.0000 | 45.977 *** 0.0070 |
| 30-November-18 | Exchange rate | Rupee depreciation from 131.95 to 136.51 | 35.194 *** 0.0000 | 16.391 0.0000 | 13.853 *** 0.0012 |
| 19-March-20 | The first outbreak of COVID-19 | First COVID-19 patient admitted to a hospital. | 12.454 *** 0.0000 | 21.840 *** 0.0000 | 24.740 *** 0.0000 |
| 23-March-20 | First lack down in Karachi due to COVID-19 | Lockdown in Karachi and closure of PSX. | 45.427 *** 0.0000 | 32.578 *** 0.0000 | 17.411 *** 0.0000 |
| 31-May-20 | Nation-wide lack down due to COVID-19 | Nation-wide lockdown due to COVID-19. | 23.004 *** 0.0000 | 31.742 *** 0.0000 | 14.900 *** 0.0000 |

The *, ** and *** are showing the significance at 10%, 5% and 1% respectively.

The performance of Pakistan stock exchange is badly affected by the political instability (Sulehri and Ali 2020; Mehmood et al. 2020) and financial events effect (Gulzar et al. 2019;

Ghouse and Khan 2017). Out of 21 political and financial events, 19 are significant and only 2 events are insignificant. This shows that the events significantly impacted the returns of stock prices of these commercial banks except the closure of News Channels, and Punjab Government opposed the "Rah-e-Najat operation" (Ghouse et al. 2021b). The results also indicate that some events have minor impact on the returns of HBL, MCB, and UBL. These events are; Nawaz Sharif became the Prime Minister of Pakistan, Civil disobedience by Imran khan, and PTI victory on 5 June 2013, 19 August 2014, and 26 July 2018, respectively. All the other events are highly significant. No study has been particularly conducted which explored the effect of these events on the performance of stock prices of commercial banks. Ghouse et al. (2021b) carried out a study on these events but it is based on Islamic banking sector.

Furthermore, Table 7 below shows the political and financial events on the stock prices volatility of HBL, MCB, and UBL.

**Table 7.** The results of impact of events on volatility of returns of banks.

| Date | Category | Detail | HBL (F-Stat) | MCB (F-Stat) | UBL (F-Stat) |
|---|---|---|---|---|---|
| 14-November-07 | Internal Affair | Closure of TV Channels | 1.8859 0.6931 | 0.4257 0.6532 | 1.4863 0.9832 |
| 19-February-08 | Elections | PPP elected their Prime Minister | 51.787 *** 0.0001 | 65.872 *** 0.0078 | 65.486 *** 0.0026 |
| 15-May-08 | Internal Affair | Lawyers Movement for Chief Justice of Pakistan (Justice Iftikhar Ahmed) | 193.450 *** 0.0005 | 16.533 *** 0.0012 | 87.562 *** 0.0018 |
| 7-September-08 | financial crisis | 2008 Global Financial crisis | 169.501 *** 0.0001 | 67.392 *** 0.0040 | 122.47 *** 0.0001 |
| 11-March-09 | Internal Affair | Long March organizaed by PML-N | 80.904 *** 0.0001 | 56.652 *** 0.0078 | 80.689 *** 0.0014 |
| 10-June-10 | Internal Affair | Punjab Government opposed Rah-e-Najaat | 129.871 * 0.0710 | 65.873 * 0.0853 | 129.877 * 0.0926 |
| 2-May-11 | Foreign Affair | Operation Neptune Spear | 84.533 *** 0.0029 | 86.765 *** 0.0016 | 81.844 *** 0.0004 |
| 28-October-11 | Long March/ Political Gathering | Long March started by PMLN | 126.084 *** 0.0000 | 83.239 *** 0.0000 | 75.476 *** 0.0000 * |
| 31-October-11 | Long March/ Political Gathering | Pakistan Tehreek-e-Insaf organized a political gathering | 176.506 *** 0.0000 | 65.846 *** 0.0000 | 81.569 *** 0.0000 |
| 18-June-12 | Internal Affair | Yousef Raza Gillani dismissal | 119.116 *** 0.0000 | 78.423 *** 0.0000 | 112.86 *** 0.0000 |
| 15-Jane-13 | Long March/ Political Gathering | Pakistan Awami Tehreek Protest against PPP | 91.389 *** 0.0006 | 84.347 *** 0.0016 | 94.847 *** 0.0003 |
| 13-May-13 | Elections | PML-N formed a new Government in Pakistan | 61.534 *** 0.0000 | 65.652 *** 0.0000 | 53.585 *** 0.0000 |
| 5-June-13 | Internal Affair | Nawaz Sharif became 20th PM of Pakistan | 52.459 *** 0.0609 | 43.742 *** 0.0785 | 86.292 *** 0.0515 |
| 15-August-14 | Long March/ Political Gathering | Protest against riggning by PAT and PTI | 196.272 *** 0.0000 | 74.652 *** 0.0000 | 74.029 *** 0.0000 |
| 19-August-14 | Internal Affair | Announcement of Civil disobedience by PTI | 146.641 *** 0.0074 | 42.237 *** 0.0052 | 55.188 *** 0.0082 |
| 20-February-18 | Internal Affair | Supreme Court terminated Nawaz Sharif on embazalment | 81.917 *** 0.0000 | 76.563 *** 0.0000 | 98.651 *** 0.0000 |

**Table 7.** *Cont.*

| Date | Category | Detail | HBL (F-Stat) | MCB (F-Stat) | UBL (F-Stat) |
|---|---|---|---|---|---|
| 13-July-18 | Internal Affair | PML-N leaders arrested by Police | 86.986 *** 0.0000 | 65.856 *** 0.0000 | 65.326 *** 0.0000 |
| 25-July-18 | Internal Affair | New Elections conducted in Pakistan | 142.906 *** 0.0000 | 78.236 *** 0.0000 | 112.48 *** 0.0000 |
| 26-July-18 | Internal Affair | PTI bagged highest seat | 136.441 *** 0.0029 | 110.72 *** 0.0051 | 37.543 *** 0.0005 |
| 9-October-18 | Exchange rate | Pakaistani rupee depreciation | 73.991 *** 0.0000 | 99.375 *** 0.0000 | 73.991 *** 0.0000 |
| 30-November-18 | Exchange rate | Rupee depreciation from 131.95 to 136.51 | 167.443 0.0000 | 87.349 *** 0.0000 | 143.35 *** 0.0000 |
| 19-March-20 | The first outbreak of COVID-19 | First COVID-19 patient admitted to a hospital. | 34.655 *** 0.0000 | 45.230 *** 0.0000 | 32.027 *** 0.0000 |
| 23-March-20 | First lack down in Karachi due to COVID-19 | Lockdown in Karachi and closure of PSX. | 65.465 *** 0.0000 | 26.264 *** 0.0000 | 32.487 *** 0.0000 |
| 31-May-20 | Nation-wide lack down due to COVID-19 | Nation-wide lockdown due to COVID-19. | 36.423 *** 0.0000 | 34.420 *** 0.0000 | 35.278 *** 0.0000 |

The * and *** are showing the significance at 10% and 1% respectively.

Table 7 shows the impact of COVID-19, political and financial events on volatility of stock prices. The results in Table 7 indicate that COVID-19 caused high volatility in the stock prices of commercial banks. All the COVID-19 events have significantly contributed to the volatility of stock prices of HBL, MCB, and UBL. COVID-19 is a cause of high fluctuations in Pakistan market prices (Shah et al. 2021). COVID-19 badly impacted the stock prices and created fluctuations in Pakistan stock exchange (Saeed et al. 2021). COVID-19 events affected the performance of Islamic banking sector (Ghouse et al. 2021b).

The performance of Pakistan stock exchange fluctuated by the political instability (Khan et al. 2022; Sulehri and Ali 2020; Mehmood et al. 2020) and financial events effect (Kashif et al. 2021; Ghouse et al. 2019). A total of 19 out of 21 political and financial events have significantly generated high volatility and only 2 events are insignificant. This shows that the events significantly impacted the volatility of stock prices of these commercial banks except the closure of News Channels, and Punjab Government opposed the "Rah-e-Najat operation". The results also indicate that some events have minor contribution in the volatility of HBL, MCB, and UBL. These events are; Nawaz Sharif became the Prime Minister of Pakistan, Civil disobedience by Imran khan, and PTI victory on 5 June 2013, 19 August 2014, and 26 July 2018, respectively. All the other events are highly significant.

The above-given results show the events that influenced commercial banks stock returns, which exist in the "KSE 100" index; only a few events did not affect the commercial banks returns. Based on the results, this study can conclude that these events (political and financial) badly affected the stock prices return of commercial banks. No study has been carried out to explore the impact of these events on the volatility of stock prices of commercial banks. Ghouse et al. (2021b) conducted a study on these events but it is based on Islamic banking sector.

## 5. Conclusions and Policy Recommendations

The economy of Pakistan is facing political instability since independence due to many factors, of which some are external, and some are internal. However, all these events affect the economy of Pakistan. Since the last decade, financial inclusion is increased in the country. Many financial dealings are done through the banking system, but any political and financial event generates shock in the economy and affects the volatile banking system

performance. Furthermore, the COVID-19 pandemic has also affected the economy badly. This study explored the effect of some significant events (political, financial and COVID-19) on the returns and volatility of commercial banks. This study found that the returns series of the stock prices of HBL, MCB, and UBL commercial banks reacts to 19 events out of 21 events.

Moreover, the volatility of HBL, MCB, and UBL commercial banks affected over 20 out of 21 events. There are two political events where the return series of the commercial banks did not show any reaction: TV Channels to go on air and Punjab Government opposed "Rah-e-Najat operation". The current study also found that only one event did not affect commercial banks' volatility, which is TV Channels to go on air on 14 November 2007.

There are two major financial events; global financial crisis, exchange rate depreciation, and these events significantly affected the returns and volatility of commercial banks. There were eighteen political events and two did not affect the returns, and one did not affect the volatility of commercial banks. There was only one political event that did not affect both the returns and volatility. Overall, commercial banks' stock prices affect financial and political events even if they are indigenous and exogenous.

Internal and extraneous shocks are the source of high fluctuation and impact the performance stock markets. The structural breaks due to unexpected events commonly changed the behavior and trend of the series. Without the identification of structural break, the valid forecasting cannot be attained. Moreover, the consistency of parameter is always questionable in presence of time variant means and variances. To tackle all these problems, it is necessary to identify the structural breaks in financial series to make a valid future predictions and sustainable policy related to financial market. The policy can be opted based on the result for commercial banks and the central bank—State Bank of Pakistan.

**Author Contributions:** Data curation, G.G.; formal analysis, G.G.; investigation, G.G. and M.I.B.; methodology, G.G. and M.I.B.; resources, M.H.S.; software, G.G.; supervision, M.I.B. All authors have read and agreed to the published version of the manuscript.

**Funding:** This research received no external funding.

**Data Availability Statement:** Not applicable.

**Conflicts of Interest:** The authors declare no conflict of interest.

## Notes

[1] For Islamic finance rview, please refer to Al Rahahleh et al. (2019) and on Islamic Mutual fund Al Rahahleh and Bhatti (2022).

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
