# Peer review of "Impact of COVID-19, Political, and Financial Events on the Performance of Commercial Banking Sector"

_jrfm, doi:10.3390/jrfm15040186_

Round 1

Reviewer 1 Report

The manuscript addresses an interesting and topical topic. The manuscript has the potential to be published but some improvements need to be made. The authors should develop a discussions section in which to comment on the results obtained in the context of similar studies that confirm or refute their own results. The conclusions section should be extended and the authors should present in full the measures proposed by the economic policy, the future research directions and the limits of the research.

Please also use this article 

Ho, L. T., & Gan, C. (2021). Foreign direct investment and world pandemic uncertainty index: Do health pandemics matter?. Journal of Risk and Financial Management14(3), 107.

Reviewer 2 Report

The paper investigates the effect of major external events (such as the Covid-19 outbreak and other relevant political and financial events) on the returns and volatility of publicly traded commercial banks in Pakistan.

Overall, the paper addresses an interesting research question but I have the following suggestions for improvements.

First, how are the major events selected? The author(s) should provide a better description of the criteria that led to the selection of those specific events, explain why they are relevant, and outline the expected effects on the stock returns of commercial banks.

Second, there might be confounding effects associated with multiple events occurring in a very short time window, e.g.: New Elections conducted in Pakistan on July 25, 2018 and PTI bagged the highest seat on July 26, 2018. How do the author(s) address this issue?

Third, I would suggest the author(s) put more effort in commenting and interpreting the results. What are the types of events that are more relevant? What are their effects on stock market valuations? Do banks react differently based on their characteristics?

Fourth, sometimes writing is poor (see, e.g., the extra “and” in the title). I recommend professional proofreading.

Round 2

Reviewer 2 Report

I would like to thank the authors for carefully addressing the comments I formulated in my previous report. I have no further points to raise.

Author Response

Thanks for your comments